# Transition from AFM Spin Canting to Spin Glass–AFM Exchange as Particle Size Decreases in LaFeO_3_

**DOI:** 10.3390/nano13101657

**Published:** 2023-05-17

**Authors:** Dhoha Alshalawi, Jose María Alonso, Angel R. Landa-Cánovas, Patricia de la Presa

**Affiliations:** 1Institute of Applied Magnetism, UCM-ADIF, A6 22,500 km, 28230 Las Rozas, Spain; dhohadho@ucm.es (D.A.); jm.a.r.0@csic.es (J.M.A.); 2Material Science Institute of Madrid, CSIC, 28049 Madrid, Spain; landa@icmm.csic.es; 3Department of Materials Physics, Complutense University of Madrid, 28040 Madrid, Spain

**Keywords:** perovskite oxides, exchange bias, nanoparticles, nanostructure

## Abstract

In this work, we have studied structural and magnetic properties of LaFeO_3_ as a function of the particle size *d*, from bulk (*d* >> 1 µm) to nanoscale (*d* ≈ 30 nm). A large number of twins were observed for large particles that disappear for small particle sizes. This could be related to the softening of the FeO_6_ distortion as particle size decreases. It was observed that the bulk sample showed spin canting that disappeared for *d* ~ 125 nm and can be associated with the smoothening of the orthorhombic distortion. On the other hand, for *d* < 60 nm, the surface/volume ratio became high and, despite the high crystallinity of the nanoparticle, a notable exchange effect bias appeared, originated by two magnetic interactions: spin glass and antiferromagnetism. This exchange bias interaction was originated by the formation of a “magnetic core–shell”: the broken bonds at the surface atoms give place to a spin glass behavior, whereas the inner atoms maintain the antiferromagnetic G-type order. The LaFeO_3_ bulk material was synthesized by the ceramic method, whereas the LaFeO_3_ nanoparticles were synthesized by the sol-gel method; the particle size was varied by annealing the samples at different temperatures. The physical properties of the materials have been investigated by XRD, HRTEM, TGA, and AC and DC magnetometry.

## 1. Introduction

The next-generation technology that overcomes the various difficulties of the current electronics (based only on electron charge) will probably be based on the degrees of freedom of the electron spin. In this sense, one of the techniques, studied with great interest, is the current-induced magnetization reversal which has been investigated mainly in metallic or semiconductor materials [1]. The flow of current, which inevitably accompanies the dissipation of energy, makes insulating materials, in which the magnetization M can be controlled with electric field, the most studied for use in new-generation electronics. The multiferroic materials are the best candidates for this goal [2] because they simultaneously present ferroelectricity, ferromagnetism, and ferroelasticity. Indeed, the multiferroic magnetoelectric materials have magnetization and dielectric polarization, which can be modulated and activated by an electric field and a magnetic field, respectively [3].

The iron perovskites (LnFeO_3_, Ln = lanthanide) have attracted much attentions for their multiferroic and magnetic–optical properties and for presenting, in general, a notable exchange bias effect [4]. All these properties are of great interest for the development of new magnetic memory devices, low-power consumption spintronics devices, and magneto-optical sensors. Therefore, it is very interesting to study the magnetic properties of LnFeO_3_, not only for scientific interests but also for applications [2,5].

LaFeO_3_ is one of the members of the orthoferrites LnFeO_3_ with perovskite structure and space group Pbnm (orthorhombic) [6]. In this material, the oxidation state of Fe is 3+ (Fe^3+^) with *d*^5^ and magnetic moment approximately 5 µB. For a bulk LaFeO_3_ sample, the magnetic moments order antiferromagnetically of G-type at room temperature and almost parallel to the *a* axis. Each Fe^3+^ ion has six first neighbors, with their moment aligned antiparallel to it (G type order) [7]. The material is antiferromagnetic (AFM) up to T_N_
≈ 750 K but with weak ferromagnetism due to spin canting (spin canting angle α ~ 9.1 mrad°) [8] with the magnetic moments points out to the *c* axis. The direction of the weak ferromagnetism is the result of the perturbing effect of the crystalline field on the exchange field, which is much stronger [9]. Surprisingly, the reduction in the particle size of these materials leads to the disappearance, sometimes, of the weak ferromagnetism which implies the vanishing of spin canting [10,11,12,13]. However, to our knowledge, this behavior has not been addressed in the literature. In other cases, the decrease in particle size produces the onset of a remarkable exchange bias, which, according to different authors [5,10,11,14], is due to the presence of a strong structural disorder on the surface. In this article, we address the problem of the vanishing of spin canting in LaFeO_3_ material by reducing the particle size; on the other side, for particle sizes *d* ≤ 60 nm, the onset of a ferromagnetic surface that interacts with the antiferromagnetic core leads to an enhanced exchange bias effect. This core-shell behavior type has no structural character but magnetic.

## 2. Materials and Methods

Single-phase nanoparticles of LaFeO_3_ were synthesized using the sol-gel Pechini method [15]. Briefly, Fe(NO_3_)_3_·9H_2_O (PanReac, 99%), La(NO_3_)_3_·6H_2_O (AlDRICH, 99.99%), citric acid (C_6_H_8_O_7_) (ALFA AESAR 99.5%), and ethylene glycol (C_2_H_6_O_2_) (ALFA AESAR 99%) were used as high-purity raw materials. In addition, 0.01 mol of Fe(NO_3_)_3_·9H_2_O and La(NO_3_)_3_·6H_2_O were gradually added to 200 mL of an 0.26 molar aqueous citric acid solution. The solution was heated at 50 °C and vigorously stirred until a colorless solution was obtained. Finally, after all the starting materials dissolved completely, 10 mL of ethylene glycol was added to the solution. The solution was evaporated to dryness, after several hours at 90 °C, forming a dark brown powder. The dray powder was then ground in an agate mill mortar and sintered at 500 °C for 24 h to remove all the remaining organic materials. Finally, the resulting powder was divided into four samples and heated in the oven for 2 h at different annealing temperatures: 600 °C (LF-600), 700 °C (LF-700), 800 °C (LF-800), and 900 °C (LF-900). 

LaFeO_3.00_ bulk (LF-Bulk) was synthesized by the conventional ceramic method, in which stoichiometric amounts of La_2_O_3_ (ALDRICH, 99.99%) and Fe_2_O_3_ (FLUKA, 99.9%) were mixed, grounded, and annealed at 1300 °C for 72 h with intermediate milling. After quenching at room temperature, a single phase was obtained. Inductively coupled plasma optical emission spectrometry (ICP-OES) was used to conduct a chemical analysis of the composition of the LF-900 and LF-Bulk samples. The material was digested in triplicate using a sulfuric/hydrogen peroxide solution. The oxygen content was determined by thermogravimetric analysis (TGA) on a Cahn D-200 electro balance by reducing samples under H_2_/He (0.2/0.3 atm.) by increasing the temperature of samples from room temperature up to 900 °C. Since the final product of the reduction process, determined by X-ray diffraction, was always the same mixture of La_2_O_3_ and Fe, the oxygen content was determined from the weight difference between the starting material and the final products. For phase identification and structural characterization of each sample, an X-ray diffraction (XRD) PANalytical X’Pert Pro MPD diffractometer with two Cu K-Alpha 1 (1.5406) and K-Alpha 2 (1.5443) radiation tubes was used at room temperature with a 2θ range from 5° to 100° and a step size of 0.03°. The XRD patterns were analyzed, and the crystal structure was determined using the Rietveld refinement technique with the Crystal Impact MATCH! [16] and FullProf package suite programs [17]. Scherrer’s formula was used to determine the average crystallite size using the X’Pert HighScore program.

To characterize the nanostructures of the sample, a transmission electron microscopy (TEM) JEOL JEM 2100 microscope at 200 keV was used to perform selected area electron diffraction (SAED). In addition, each sample composition was analyzed using X-ray energy-dispersive spectroscopy (XEDS) and microanalysis equipment (Oxford INCA). High-Resolution Electron Microscopy (HRTEM) images were collected using a JEOL JEM-3000F transmission electron microscope with an acceleration voltage of 300 kV and a structural resolution of 1.7 for samples LF-600, LF-900, and LF-Bulk. High-resolution images at different values of thickness and defocus were simulated using the TempasX software package (Total Resolution LLC) and the multislice calculation method. 

The zero field-cooled (ZFC) hysteresis cycles M-H of each sample were measured at 5 K and 300 K and 50 kOe applied field using a Quantum Design SQUID magnetometer. Field-cooled (FC) hysteresis cycles were measured by cooling down at 50 kOe and measuring at 5 K and 50 kOe maximum applied field for samples LF-600 and LF-700. Thermal dependence of the magnetization was measured at 1000 Oe applied magnetic field under ZFC–FC procedures from 5 K to 300 K for samples LF-600, LF-700, LF-800, and LF-900. Using the commercial PPMS platform Quantum Design, AC susceptibility (χac) temperature dependency characterization was performed for samples LF-600 throughout the temperature range from 2 K to 200 K with 4 Oe and frequencies of 10, 100, 1000, and 10,000 Hz. 

## 3. Results

### 3.1. Compositional Analysis and Structural Characterization

XEDS analyses in LF-600 samples show an average composition of La_1.09_Fe_0.91_O_3_, which fits well with the nominal one when experimental errors (~10%) are considered. Since all the samples are synthesized from the same nanoparticle batch (the sol-gel), they all have the same stoichiometry and composition. In addition, the samples are homogeneous without the presence of starting oxides or secondary phases (See Appendix A). 

In order to obtain a more accurate composition, ICP analysis of the materials was carried out, and the value obtained was La_1.01_Fe_1.00_O_3_, the same as the nominal composition.

### 3.2. Thermogravimetric

The thermogravimetric (TG) results are shown in Figure 1 for samples LF-600, LF-900 and LF-Bulk. From room temperature up to 450 °C, the samples suffer a small weight loss of ~1.25% for LF-600 and ~0.90% for LF-900, which corresponds to the removal of water and residual organic matter from the sol-gel process [13] (see Figure 1a). Unsurprisingly, the weight loss is smaller for the sample treated at a higher temperature (see Figure 1b). Between temperatures 450 °C and 750 °C, the weight loss of the materials is very small, just 0.1%, in the stability zone of the LaFeO_3_. Above 750 °C, the samples suffer a strong weight loss of approximately 10 %wt, which corresponds to the weight loss associated with the reduction reaction of the material (reaction (1)). This behavior confirms that the oxygen stoichiometry of these materials is 3.00 ± 0.01.
(1)$${\mathrm{LaFeO}}_{3}+3/2H_{2}\to 1/2{\mathrm{La}}_{2}O_{3}+\mathrm{Fe}+3/2H_{2}O$$

The bulk sample shows different behavior, as shown in Figure 1b. This material is stable between room temperature and 750 °C without any weight loss. From this temperature, there is a loss of weight (~10%) corresponding to the total reduction of LaFeO_3_ material (reaction (1)). It is worth noting that the weight loss of the bulk material has slower kinetics than sol-gel samples due to the larger particle size of the ceramic sample. 

### 3.3. X-Ray Diffraction (XRD) Patterns

XRD patterns of samples of LF-600 nanoparticles and LF-Bulk recorded at room temperature (see Figure 2) show single-phase, well-defined patterns without additional impurities of the starting materials or secondary phases. The Rietveld refinement has been achieved using high-resolution XRD. All maxima were assigned to the LaFeO_3_ phase with orthorhombic symmetry and Pnma spatial group, where lattice parameters were calculated, as shown in Table 1, for all samples.

The nanoparticle size was determined from XRD using the Scherrer equation. For sol-gel samples, the nanoparticle size smoothly increases from 27 nm to 125 nm when the annealing temperature increases (see Table 1 and Appendix A). It is worth noting the small orthorhombic distortion of the LaFeO_3_ in bulk, with a/c = 1.002, similar to values previously reported [18]. In addition, the a/c decreases with decreasing particle size, as expected. Furthermore, if the orthorhombic parameters are transformed to the pseudo cubic ones, with a_c_ = a_o_/√2, b_c_ = b_o_/2, and c_c_ = c_o_/√2, then c_c_ = b_c_ for the bulk (see Appendix A). 

### 3.4. High-Resolution Transmission Electron Microscopy (HRTEM)

The particles observed by TEM present particle sizes in the range measured by X-ray diffraction. At low annealing temperatures (LF-600 and LF-700 samples), the particles that constitute the material have a rounded shape forming nearly spherical aggregates (see Appendix A). Appendix A shows the particle size distribution for LF-700, with a mean particle size d = 29 nm, which virtually matches the values obtained by Scherrer’s formula. 

A high-resolution image of a typical nanocrystal of sample LF-600 with a diameter of approximately 14 nm is shown in Figure 3. The high crystallinity of the samples treated at the lowest temperatures is noteworthy. The distance between planes is d = 3.93 Å, which corresponds to the (002) planes of LaFeO_3_ perovskite. The (200) atomic planes with d = 2.87 Å are also indicated. They are more difficult to see in the image due to their smaller spacing, but the FFT shows that they are there. They can be more easily observed at the edge of the nanoparticle since thinner parts of the crystal always provide a higher image resolution. The corresponding fast Fourier transform (FFT) is shown in Figure 3b and indicates that the nanocrystal is orientated along the [010] zone axis. The observed diffraction maxima agree with the Pbnm space group (S.G. 62): the forbidden reflections h00; h= 2n + 1 and 00l; l = 2n + 1 do not appear. The h0l: h + l = 2n + 1 reflections are also forbidden due to the n-glide plane perpendicular to *b*. When a glide plane is perpendicular to the observed zone axis, as such is the case, there are no allowed paths to add intensity to the forbidden maxima through multiple diffraction. It is important to note that LaFeO_3_ nanoparticles do not constitute a core–shell system from a structural point of view and they show very clean and crystalline surfaces. 

Figure 4 shows TEM results for sample LF-900, where the beginning of the sintering process can be observed. Three different particles, with diameters greater than 100 nm, share a common triple point and three different border grains separate them, although in some areas they are still not joined (Figure 4a). Two of the particles are related by a twin operation but the third one is not orientated. The twin boundary seems to be very crystalline (see Figure 4b). The FFTs shown in Figure 4c,d correspond to selected areas from both twins, marked as white squares in Figure 4b. Both twins are orientated along the [-101] zone axis but they are tilted 90 degrees on the plane. There is a big difference from both FFTs: the one in Figure 4c shows the forbidden reflections 0k0: k = 2n + 1, while the one in Figure 4d does not. The reason is the different thickness of both twins, with the one shown in Figure 4c being the thickest. The higher thickness implies multiple diffraction phenomena that transfer intensity to the forbidden reflections. For example, an electron scattered along the 212 reflection can later be scattered along the -20-2 reflection, if the crystal is thick enough, and will end on the 010 forbidden reflection.

This difference in thickness also gives rise to the different contrast shown by the two twins in the HRTEM image. The twin boundary lies along the [111] direction, which is common to both twins. 

We also used TEM to check the LaFeO_3_ bulk sample (LF-Bulk). We observed a great number of twins in the LaFeO_3_ crystals (see Figure 5). In the small area covered by the TEM image, ca. 50 nm × 30 nm^2^, we found four different twinned domains. We obtained FFT patterns from each of the small domains shown in the HRTEM image: #1, #2, #3, and #4. 

One of the domains (#1) is in [001] orientation, while the other three are in two different [100] orientations, named as [100]′ (#3) and [100]″ (#2 and #4). The last two twinned domains are tilted 90° on the plane with respect to #3 domain, although all of them are oriented along the [100] zone axis. 

It can also be considered that both the #2 and #4 twinned domains are the same domain since they are contiguous, but it can be noted in Figure 6 that the #2 domain disappears under the electron beam. 

It is worth noting that the size of the smallest domains is of the same range as the nanoparticles of sample LF-600. These small domains seem to be rather unstable since after a couple of minutes under the electron beam at 200 kV they disappear from HRTEM images (see Figure 6). The cause for this modification of structure can be due to radiation damage, but this usually shows amorphized edges, which is not the case here. 

Another possibility is that the heating of the crystal by the electron beam produces the atomic rearrangement of the smallest twins. This is probably due to a mixing of both cases, but it is difficult to demonstrate it. 

It is important to note that the nanoparticles of sample LF-600 are perfectly ordered, while the LF-Bulk sample presents a large number of twins. This is probably due to the bigger distortion of the unit cell in the bulk sample. Sample LF-900 seems to be an intermediate case with twins but with not as many as the bulk sample.

### 3.5. Magnetic Properties

The bulk material shows, from 50 kOe to 20 kOe, a linear behavior of the magnetization corresponding with the AFM character of the magnetic interactions (see Figure 7). However, for H < 15 kOe, the linear behavior is replaced by a hysteresis cycle due to the spin canting presented by the orthoferrites [9,19]. This spin canting is the result of the disturbing effect of the crystalline field on the much stronger exchange field [9]. By fitting the magnetization at high field (1.5 kOe ≤ H ≤ 50 kOe) with a lineal function, the saturation magnetization of the spin canting can be estimated as M_s_ = 2.23 emu/g (0.097 μ_B_). If we consider that Fe^3+^ has a magnetic moment of 5 μ_B_ and the saturation magnetization of LaFeO_3_ bulk is 0.097 μ_B_, it is possible to calculate the canting angle at α = 19 mrad (see Appendix A). 

Figure 7 also shows a comparison of the magnetic behavior between bulk and large nanoparticles (LF-900; d ≈ 125 nm). It is worth noting that the spin canting has disappeared in LF-900, whereas the slope of the magnetization curve is the same as the high-field magnetization of the bulk sample. These results suggest that, as the particle size is reduced, only the AFM interaction survives and the spin canting disappears because the crystalline field has no longer effect on the AFM interactions. This behavior has been already observed by Manchon-Gordon et al. [20].

These results are consistent with those observed by XRD and HRTEM. As previously mentioned, for bulk material, a/c = 1.002 is higher than for nanoparticles, a/c = 1.0001 (for LF-700), indicating a decreasing orthorhombic distortion as particle size decreases. Therefore, if the octahedral distortion of Fe^3+^ decreases, the crystalline field weakens and has a smaller effect on the AFM exchange interaction. It is worth noting that, according to HRTEM images, the nanoparticles are perfectly ordered without the core–shell, amorphous phase, or twins, whereas the crystals of the bulk sample show a lot of twins. On the other hand, as the particle size decreases, a new ferromagnetic interaction appears that becomes more important with decreasing particle size (Figure 8). As is well known, the surface to volume ratio (S/V) increases as particle size decreases, making the contribution of the surface atoms more relevant to the total magnetization. When the magnetization of the ferromagnetic phase is plotted against S/V, it is observed that the ferromagnetic fraction increases exponentially with S/V (see Figure 9). This behavior suggests that the onset of ferromagnetism with decreasing particle size is given by the contribution of the surface atoms to the whole magnetization, as previously reported by other authors [21].

The existence of ferromagnetic behavior in orthoferrite nanoparticles has been previously reported by other authors and associated with broken bonds at the surface atoms [4,10,13]. Besides the broken bonds, the lacking oxygen at the surface cells induces changes in the iron oxidation state in order to maintain electrical neutrality [21]. In fact, the nanoparticle surface is an ideal place for the onset of magnetic disorder and frustration. Indeed, the study of the magnetization as a function of temperature under an applied field of 1 kOe (Figure 10) shows that ZFC–FC curves have different behaviors in the low temperature region (T < 100 K). 

The ZFC curves present an increasing magnetization with the temperature until reaching a maximum. The temperature of the maximum magnetization decreases as the annealing temperature increases, i.e., with increasing particle size. The FC curves present two different behaviors at low temperatures: in the case of LF-600 and LF-700, the magnetization is almost constant at low temperatures and then decreases, whereas for LF-800 and LF-900, the magnetization always decreases with increasing temperatures. Such behavior is characteristic of the existence of a strong magnetic disorder due to the presence of frozen moments, which may be indicative of the formation of a spin glass (SG). However, a superparamagnetic behavior (SPM) of the nanoparticles can also lead to a marked difference between the ZFC and FC curves. The fact that the temperature of the maximum that appears in the ZFC curves shifts towards lower temperatures as the particle size increases suggests the prevalence of spin glass-like behavior [22].

To understand the nature of our system (SG or SPM), AC magnetic susceptibility has been measured. Figure 11 shows the temperature dependence of the real part of the AC magnetic susceptibility of LF-600 at different frequencies in the range of 10–10,000 Hz and 4 Oe. The data for $\chi^{\prime }\left(T\right)$ show the expected behavior of SPM/SG systems, i.e., both show a peak T ≈ 17 K and the peak positions are frequency dependent. To identify the nature of the LF-600 nanoparticles, the real part of AC susceptibility ($\chi^{\prime }\left(T\right)$) is used through empirical parameter ϕ = ΔT_m_/(T_m_Δlog_10_(f)), which represents the relative shift of the T_m_ peak vs. frequency. T_m_ represents the average blocking temperature observed across the range of frequencies tested in the experiment. ΔT_m_ is the difference between T_m_ measured at the Δlog_10_(f) frequency interval. This parameter provides a good criterion for distinguishing SPM from SG. For superparamagnetic systems, ϕ ~ 0.1, while for spin glass systems, ϕ < 0.05 [23]. In our case, ϕ≈ 0.029, which confirms that in our system there is an SG, with a freezing temperature of Tf ~ 170 K. 

All these results suggest that the nanoparticles synthesized by sol-gel have a core–shell structure, with an AFM core and a spin glass shell. 

It is well known that the frozen state of the spin glass depends on the previous history, in particular, on the applied field that the sample has been cooled down to [22,24]. Figure 12 shows the hysteresis cycles of nanoparticles LF-600 and LF-700 cooled down to 5 K at different applied fields, with H_cooling_ = 0, 5, 10, 20, and 50 kOe. From these results, it is possible to observe that the hysteresis cycles are open, which is a characteristic behavior of a disordered system with a spin glass phase [10,25]. In addition, there is a noticeable exchange bias effect. The coercive field (H_C_) and the exchange bias field (H_EB_) increase with the cooling field (see Table 2). 

The exchange bias originated from the coupling of the spin glass with the AFM phase. In this case, the two coexisting magnetic phases are the AFM core and the surface spin glass. The exchange bias between AFM–SG is not very common, although there are a few examples [21]. Recently, Maniv et al. [24] reported an exchange bias between AFM and SG in Fe_x_NbS_2_, with the exchange bias field higher than 10 kOe. 

In our case, the exchange bias in the LF-700 sample is especially significant, when, after cooling with a 50 kOe field, the H_EB_ is notably higher than the H_c_. 

## 4. Discussion

The main results obtained can be summarized as follows:(1)According to HRTEM, the LaFeO3 in bulk has a large number of twins and the magnetic behavior corresponds to the well-known AFM with spin canting.(2)For particle size *d ~* 125 nm, the number of twins decreases and the spin canting vanishes completely.(3)For *d* < 60 nm, a new ferromagnetic interaction appears that becomes more important as particle size decreases.

Figure 13 summarizes these main milestones. This figure shows the variation of the surface-to-volume ratio (S/V) of the nanoparticles size represented as a function of their radius. The red dots represent the radii of the nanoparticles in this article. 

For *r* < 20 nm, the nanoparticles present a spherical morphology with high crystallinity and a marked monocrystalline character. However, given the high S/V ratio, the surface properties have a marked influence on the general properties of the nanoparticles. Even though the HRTEM images show highly crystalline nanoparticles, this does not prevent that broken bonds at the nanoparticle surface lead to the onset of highly frustrated magnetic moments at the surface atoms. Consequently, a core–shell structure appears which has no structural character but magnetic [21]: the inner atoms remains AFM in the core, whereas the highly frustrated magnetic moments at the surface lead to SG behavior, as confirmed by AC susceptibility characterization. The coexistence of AFM and SG behavior is proved by measuring hysteresis cycles under the FC condition: the coupling between the SG and the AFM order causes a notable exchange bias effect in the M vs. H, where values can be as high as 2 kOe.

As the particle size grows, the S/V decreases and the effect of the surface on the overall properties of the nanoparticles is less noticeable. For *r* > 20 nm, the SG character of the surface is no longer appreciated and only the AFM order of the core of the nanoparticles is observed. For *r* ~ 60 nm, the hysteresis curves show, at high field, the marked linear dependence which is characteristic of AFM materials, indicating that neither spin canting nor spin glass are present at this particle size [20]. The spin canting is the consequence of the effect of the crystalline field on the exchange field, and it is known that the more distorted the FeO_6_ octahedra are, the stronger the crystalline field is. The XRD data confirm the a/c ratio with decreasing particle size, suggesting that the FeO_6_ octahedra becomes less distorted. When the FeO_6_ octahedra becomes less distorted, the effect of the crystalline field on the much stronger exchange field vanishes, making the spin canting disappear. 

As the size of the particles further increases, twins begin to appear. The bulk material, with a particle size of microns, presents a high number of twins and an AFM order with a notable spin canting, which gives rise to M vs. H curves with marked hysteresis. The onset of the spin canting is also associated with the higher distortion of the orthorhombic structure which gives rise to a stronger crystalline field affecting the exchange field [8].

## 5. Conclusions

The structural and magnetic properties in LaFeO3 have been investigated as a function of the particle size, from the bulk (particle size larger than micrometers) to the nanoscale (particle size of approximately 30 nm). HRTEM analysis showed that the bulk material contains a significant number of twins that tend to disappear for decreasing particle sizes. These twins are probably originated by the large distortion of the FeO_6_ octahedra, and they vanish because the FeO_6_ octahedra becomes less distorted as particle size decreases.

For particle size ~ 125 nm, spin canting observed in the bulk material vanishes and it is related to the decrease of the orthorhombic distortion.

As particle size further decreases, ferromagnetic behavior appears. The onset of this ferromagnetism at a small particle size is probably due to broken surface bonds and changes in the oxidation state of surface irons that result in SG behavior. On the other hand, the inner part of the nanoparticles remains AFM. Therefore, the nanoparticles constitute a core–shell magnetic structure and the coupling between the AFM core and SG shell leads to a notable exchange bias effect where the Heb field can reach a value as high as 2 kOe.

## Figures and Tables

**Figure 1 nanomaterials-13-01657-f001:**
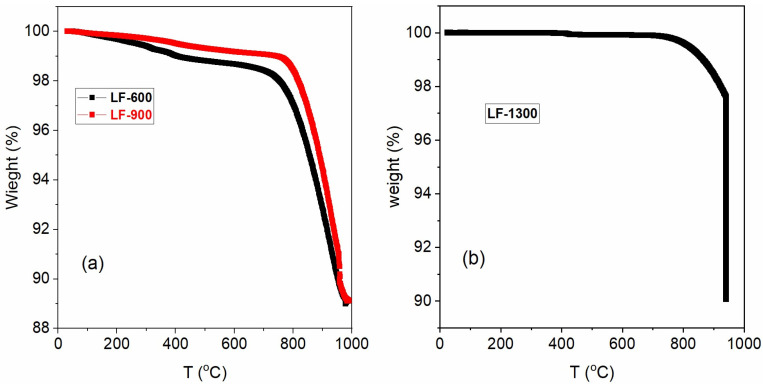
Thermogravimetric results of weight loss (left axis) and the temperature variation for samples (**a**) LF-600 (black line) and LF-900 (red line) (**b**) for LF-Bulk.

**Figure 2 nanomaterials-13-01657-f002:**
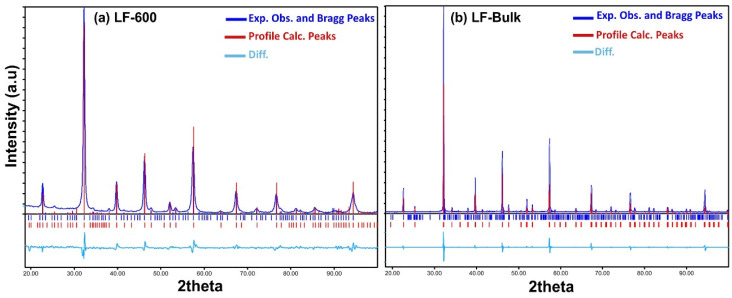
XRD experimental patterns for samples (**a**) LF-600 and (**b**) LF-Bulk. The blue line indicates the Bragg peak positions and the experimental observed pattern, while the red line indicates the profile reference LaFeO_3_ position.

**Figure 3 nanomaterials-13-01657-f003:**
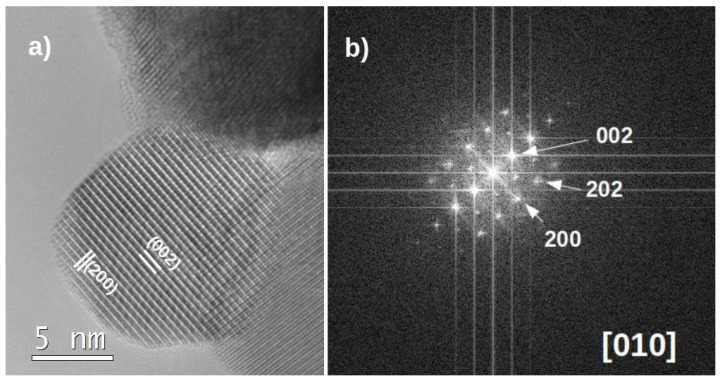
(**a**) High-resolution TEM image (HRTEM) of a nanocrystal from sample LF-600 orientated along the [010] zone axis. (**b**) FFT obtained from the HRTEM image of Figure 3a.

**Figure 4 nanomaterials-13-01657-f004:**
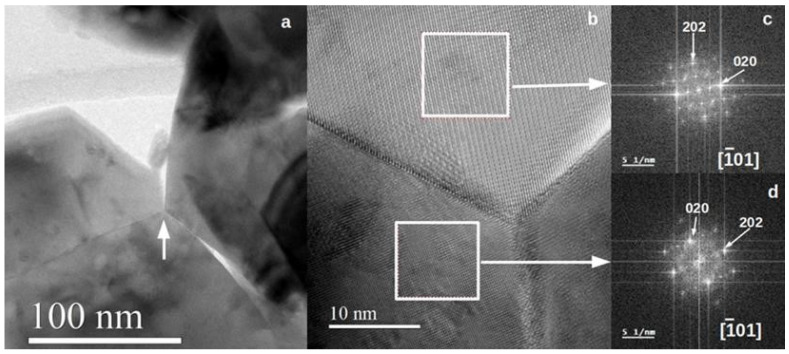
(**a**) Low magnification image from sample LF-900 showing a triple common point (indicated with an arrow) between three different particles. (**b**) HRTEM image of the triple point. The two square-marked areas correspond to the FFT diagrams shown in (**c**) and (**d**).

**Figure 5 nanomaterials-13-01657-f005:**
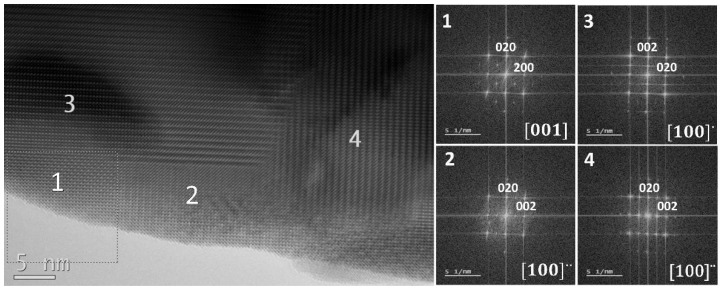
HRTEM image of a crystal from bulk sample showing four different twins. The fast Fourier transforms (FFT) of the different domains have been noted, with numbers 1, 2, 3, and 4 corresponding to the domains indicated. The size of the window used to obtain the FFTs is shown in the down-left corner of the image.

**Figure 6 nanomaterials-13-01657-f006:**
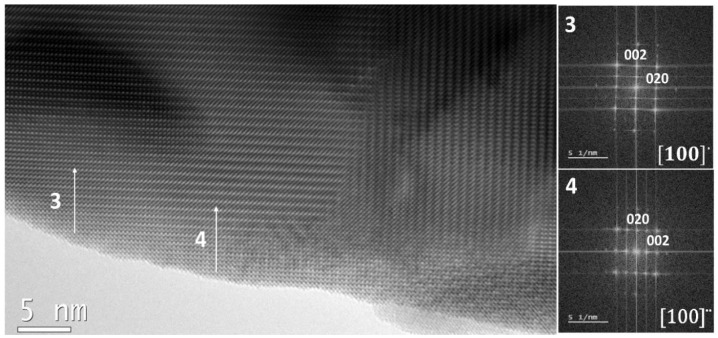
HRTEM image from the same LaFeO_3_ crystal as in Figure 5 taken a few seconds later. White arrows indicate the place where the two small twins were before radiation damage eliminated them (see Figure 5). The FFT are inserted in the upper corners of the top of their corresponding twins.

**Figure 7 nanomaterials-13-01657-f007:**
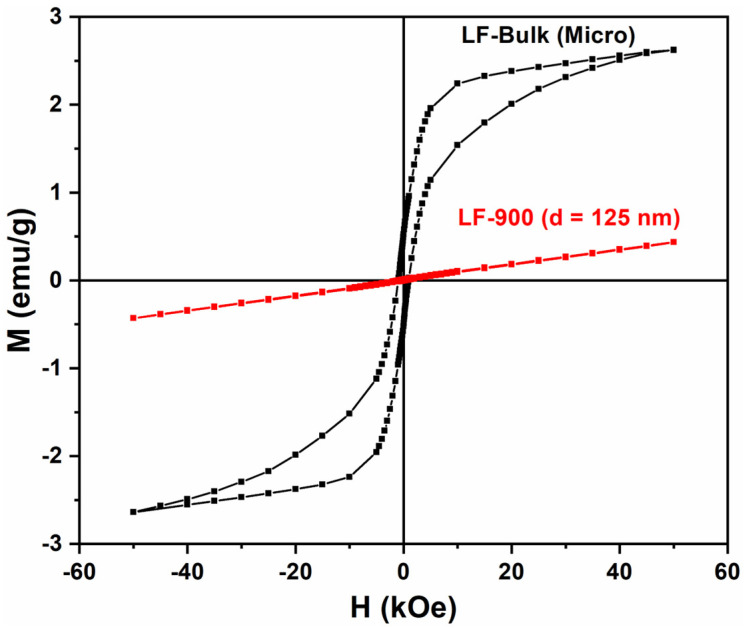
The hysteresis M-H curves of samples LF-Bulk (black curve) and LF-900 (red curve) at 5 K.

**Figure 8 nanomaterials-13-01657-f008:**
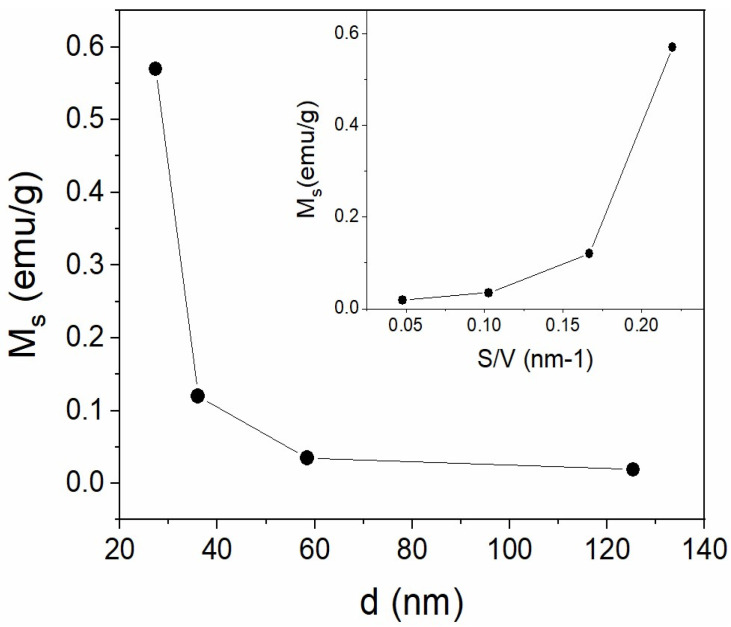
Saturation magnetization M_s_ vs. particle size d. The inset shows M_s_ vs. the ratio S/V.

**Figure 9 nanomaterials-13-01657-f009:**
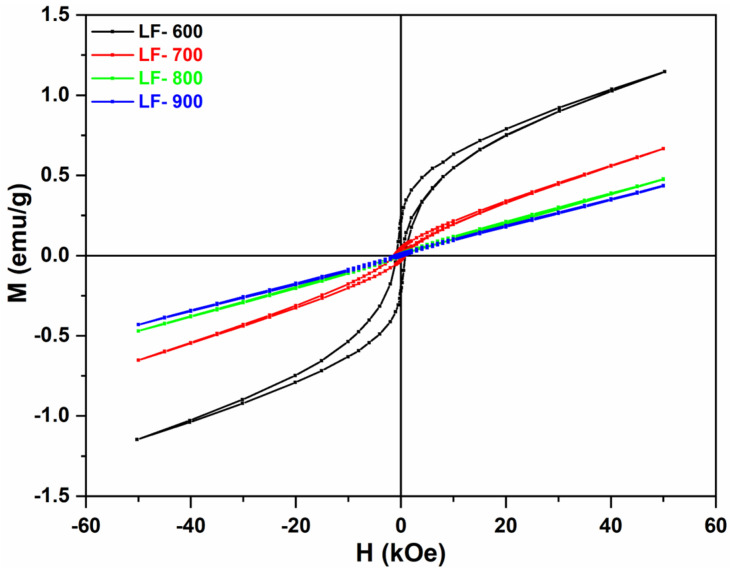
Hysteresis curves at 5 K and 50 kOe, with the experimental curves shown for samples LF-600 (black), LF-700 (red), LF-800 (light green), and LF-900 (blue).

**Figure 10 nanomaterials-13-01657-f010:**
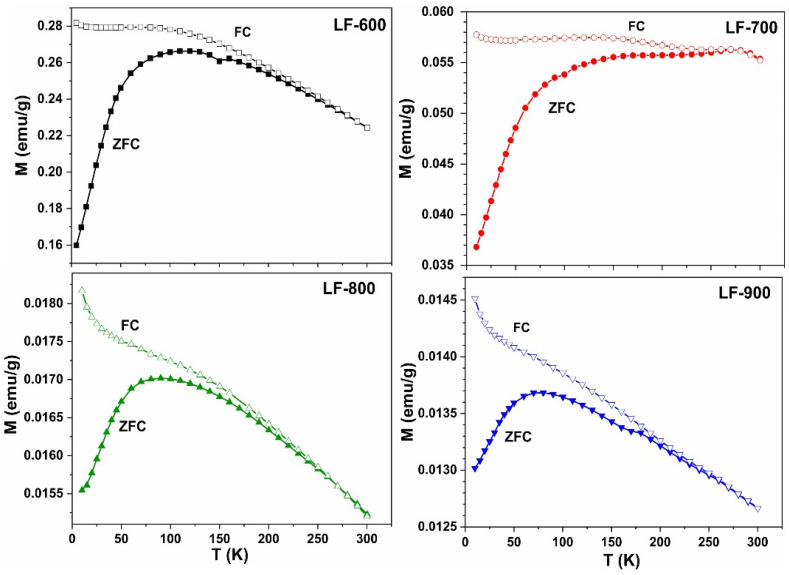
Separated ZFC–FC curves that were measured at 1000 Oe in temperatures range 10 K–300 K. Hollow marks show the FC curves, while the full marks show the ZFC curves. ZFC–FC curves are shown for the samples LF-600 (black line), LF-700 (red line), LF-800 (green line), and LF-900 (blue line).

**Figure 11 nanomaterials-13-01657-f011:**
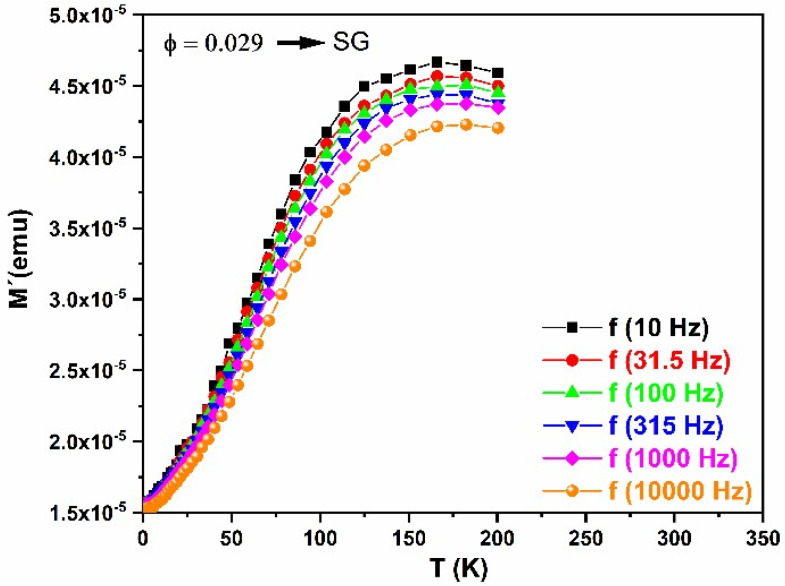
Magnetic susceptibility AC measurement of sample LF-600 at AC magnetic field 4 Oe with different frequencies in the range of 10–10,000 Hz.

**Figure 12 nanomaterials-13-01657-f012:**
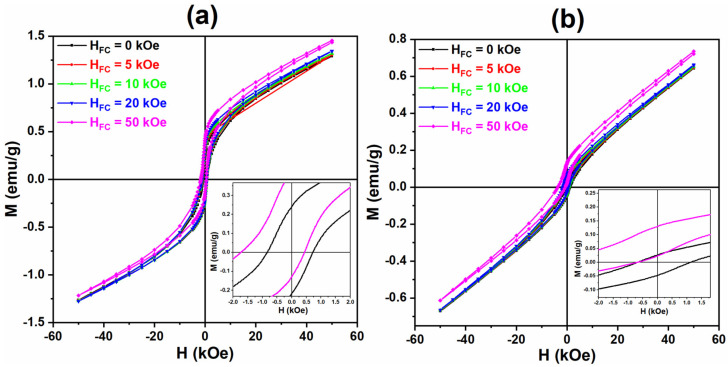
The M-H curves of different applied cooling fields from 0 to 50 kOe of samples (**a**) LF-600 and (**b**) LF-700.

**Figure 13 nanomaterials-13-01657-f013:**
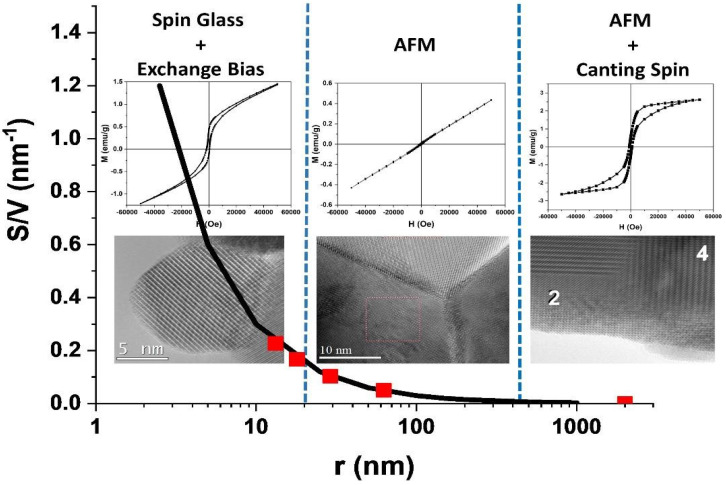
The relationship of the magnetic behavior (hysteresis curves) with the morphology of the particles (HRTEM images) and the ratio S/V (black curve and red points).

**Table 1 nanomaterials-13-01657-t001:** Rietveld refinement parameters, cell volume, and determined nanoparticle size *d* by Scherrer equation using XRD results for samples LF-600, LF-700, LF-800, LF-900, and LF-Bulk. The errors are in parentheses.

Samples	Cell Parameters			V (Å^3^)	d (nm)
	a (Å)	b (Å)	c (Å)		
LF-600	5.56338 (0.00108)	7.85760 (0.00250)	5.55924 (0.00190)	243.021 (0.123)	27
LF-700	5.56443 (0.00058)	7.85806 (0.00137)	5.55911 (0.00096)	243.076 (0.065)	36
LF-800	5.56476 (0.00034)	7.85676 (0.00059)	5.56005 (0.00041)	243.091 (0.030)	58
LF-900	5.56532 (0.00020)	7.85696 (0.00034)	5.55926 (0.00024)	243.087 (0.017)	125
LF-Bulk	5.56591 (0.00016)	7.85406 (0.00026)	5.55416 (0.00019)	242.800 (0.013)	____

**Table 2 nanomaterials-13-01657-t002:** The data of exchange bias field (H_EB_), coercive field (H_C_), and cooling field (H_FC_) of samples LF-600 and LF-700.

	LF-600		LF-700	
H_FC_ (kOe)	H_C_ (Oe)	H_EB_ (Oe)	H_C_ (Oe)	H_EB_ (Oe)
0	830	0	894	0
5	894	−255	985	−229
10	910	−296	1016	−350
20	945	−344	1074	−502
50	1077	−635	1483	−2157

## Data Availability

There is no data availability.

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
