# Peer review of "Transition from AFM Spin Canting to Spin Glass–AFM Exchange as Particle Size Decreases in LaFeO3"

_nanomaterials, 2023, doi:10.3390/nano13101657_

Round 1

Reviewer 1 Report

The work entitled “Transition from AFM spin canting to spin glass-AFM exchange as size decreases in LaFeO3” is an interesting study and a good match with this journal.

The authors obtained a number of interesting results, in particular, the disappearance of small twin domains under application of the electron beam. The authors quite convincingly show that the nature of the ferromagnetic behavior of the small particle is the formation of a spin glass, although there are data that for nano-LaFeO3 the M(H) curve can be well fitted by Langevin function for superparamagnetic nanoparticles (Nguyen Thi Thuy and Dang Le Minh, Adv. Mater. Sci. Eng. Vol. 2012, Article ID 380306, doi:10.1155/2012/380306).

At the same time, the interpretation of the magnetic properties of for LaFeO3 in the bulk state raises a number of questions.

The hysteresis loop at H < 1.5 T for LaFeO3 in the bulk state is attributed by the authors to the spin canting. However, it is known that the existence of two magnetic sublattices (soft and hard magnetic sublattices) is possible in the orthoferrites while maintaining the structural single phase of the sample. The metastable small twin domains observed by the authors may be the basis for such a soft magnetic phase. In addition, there is evidence that for high-quality LaFeO3 samples this hysteresis loop at the low fields is absent (Alejandro F. Manchón-Gordón et al, Materials 2023, 16, 1019. https://doi.org/10.3390/ma16031019).

Two technical comments:

1) The authors use both Oersted and Tesla as units for measuring the magnetic field. However, in the scientific literature it is customary to use only one of the measurement systems.

2) In the literature on the magnetic properties of the materials, the term “domains” is usually used to describe “magnetic domains”. It is better for authors to use something else instead of “twin domains”, for example, “twin boundary between the neighboring grains” or “twin structure”.

Work requires a minor revision.

Author Response

The answers are in the attached file.

Reviewer 2 Report

The article entitled “Transition from AFM spin canting to spin glass-AFM exchange as size decreases in LaFeO3” by D. Alshalawi with coauthors is devoted to investigation of a vanishing of the spin canting of the bulk LaFeO3 material with the decrease of the particle size. The authors prepared a series of the materials with different sizes and investigated in details their crystallinity by XRD and HR-TEM methods. Also, the authors investigated magnetic of the materials obtained and properties and compare it with crystallinity data. In my opinion, the results discussed in this manuscript fully explained the reason for the phenomenon under study and this manuscript can be published in the Nanomaterials. However, before publication the authors should provide Supplementary materials for revision (I didn’t find it in the submission). Also, the Equation 1 is incorrect (not it is not equation, but scheme). The authors call this process as reduction but there is no any reduction reagent. Obviously, the TG were performed under H2/He atmosphere (but not H/He atmosphere, it should be corrected too) and H2 acts as a reductant. This, should be added to the reaction equation.

Author Response

First of all, we´d like to acknowledge the reviewers for their valuable comments, suggestion, and the time dedicated for this analysis which evidently have helped to improve this work. In follows, we answer point by point to the reviewer.

The article entitled “Transition from AFM spin canting to spin glass-AFM exchange as size decreases in LaFeO3” by D. Alshalawi with coauthors is devoted to investigation of a vanishing of the spin canting of the bulk LaFeO3 material with the decrease of the particle size. The authors prepared a series of the materials with different sizes and investigated in details their crystallinity by XRD and HR-TEM methods. Also, the authors investigated magnetic of the materials obtained and properties and compare it with crystallinity data. In my opinion, the results discussed in this manuscript fully explained the reason for the phenomenon under study and this manuscript can be published in the Nanomaterials. However, before publication the authors should provide Supplementary materials for revision (I didn’t find it in the submission). Also, the Equation 1 is incorrect (not it is not equation, but scheme). The authors call this process as reduction but there is no any reduction reagent. Obviously, the TG were performed under H2/He atmosphere (but not H/He atmosphere, it should be corrected too) and H2 acts as a reductant. This, should be added to the reaction equation.

The text as well as the reaction have been corrected. Supplementary Information has been uploaded.

Reviewer 3 Report

This work is devoted to a size-change driven transition from spin canting to spin glass-AFM exchange in LaFeO3 nanoprticles below d = 125 nm. The samples of LaFeO3 bulk was synthesized by ceramic method, the nanoparticle samples LF-600, LF-700, LF-800, and LF-900 obtained by  sol-gel method. The characterization, structural and magnetic measurements are discussed in detail. It is shown that spin canting in LaFeO3 material is vanished for the particles with the reduces size.  The results are scientifically sound, the manuscript is well organized. However, this manuscript requires a revision:

1. Sections 4. Discussion and 5. Conclusions are too short. Both do not even mention the compound at all. These sections should be extended to be supported by the results. E.g., the less distorted FeO6 octahedra for the decreased particle size are proposed based on indirect observations. In lines 231-232, the fact that the LF-bulk sample presents a big amount of twins is probably due to the bigger distortion of the unit cell in the bulk sample. It is further related to the hysteresis M-H curves discussion, etc. The distortions are speculated about, no atomic coordinates and angles are measured or discussed.

2. Abstract and Conclusions belong to two different papers.

3. In Section 4, a comparison with the literature should be provided, as well as prospects of this work. 

4.  Ref. 7 should be checked and corrected: Wollan, E.O.K., W. C. , 

5. Refs. 8, 20 and 21 are given without the journal title.

6. The supporting information is not provided for review.

After these minor changes the manuscript can be published in Nanomaterials. 

Minor English correction might be needed

Author Response

THe answers are in the attached file.

Round 2

Reviewer 2 Report

The authors gave adequate comments and made requested corrections. The supplementary information presented is fine. In my opinion, now the article can be published in its current form.